# Hearing Aid in Vestibular-Schwannoma-Related Hearing Loss: A Review

**Valerio Maria Di Pasquale Fiasca** [1,*,†], **Flavia Sorrentino** [1,†], **Martina Conti** [1], **Giulia De Lucia** [1], **Patrizia Trevisi** [1], **Cosimo de Filippis** [2], **Elisabetta Zanoletti** [1] **and Davide Brotto** [1]

1 Section of Otolaryngology, Otolaryngology Unit, Department of Neurosciences, University of Padova, Via 5 Giustiniani 2, 35128 Padua, Italy
2 Audiology Unit, Department of Neuroscience DNS, University of Padova, 31100 Treviso, Italy
* Correspondence: valeriomaria.dipasqualefiasca@studenti.unipd.it
† These authors contributed equally to this work.

**Abstract:** (1) Background: Several types of hearing aids are available for the rehabilitation of vestibular-schwannoma (VS)-related hearing loss. There is a lack of recently published papers regarding this theme. The aim of the present work is to organize current knowledge. (2) Methods: A review of the literature regarding the topics "vestibular schwannoma", "hearing loss", and "hearing aid" was performed. Nineteen studies were thus considered. (3) Results: Conventional hearing aids, contralateral routing of signal (CROS) aids, bone anchored hearing aids (BAHA), and others are available options for hearing rehabilitation in VS patients. The speech discrimination score (SDS) is considered the best measure to assess candidacy for rehabilitation with hearing aids. The best hearing rehabilitative conditions in VS patients when using conventional hearing aid devices are a mild−moderate hearing loss degree with good word recognition (more than 50% SDS). CROS-Aid and BAHA are reported to be beneficial. CROS-Aid expands on the area of receiving hearing. BAHA aids use direct bone-conduction stimulation. Unfortunately, there are no available studies focused specifically on VS patients that compare CROS and BAHA technologies. (4) Conclusions: Hearing aids, CROS, and BAHA are viable options for rehabilitating hearing impairment in VS, but require an accurate case-by-case audiological evaluation for rehabilitating hearing impairment in VS. Further studies are needed to prove if what is currently known about similar hearing illnesses can be confirmed, particularly in the case of VS.

**Keywords:** vestibular schwannoma; hearing loss; hearing rehabilitation; hearing aid; CROS; BiCROS; BAHA





## 1. Introduction

Vestibular schwannoma (VS), or acoustic neuroma, is a benign Schwann-cell-derived tumor that originates from the glial layer enveloping the VIII cranial nerve. VS most frequently develops from the vestibular branch of the cochlear nerve (CN) [1]. It is mostly sporadic (95%), single, and unilateral. It is possible to find multiple and bilateral VS in patients affected by genetic syndromes such as neurofibromatosis type 2 (NF2; 5%) [2].

The possible management choices depend on the tumor and on the patient's hearing features. The therapeutic options are observation, radiotherapy, or surgery. According to size and preoperative hearing, translabyrinthine and hearing-preservation approaches (retrosigmoid and middle cranial fossa) are the most frequently performed. In selected conditions, hearing preservation surgery (HPS) represents the best therapeutic choice [3,4]. Observing VS with periodic imaging scanning, such as magnetic resonance imaging (MRI), associated with audiological assessments is possible for the management of small and asymptomatic VS. This treatment has the aim of monitoring tumor growth and hearing function before deciding to start a possible active therapy. Radiotherapy is an available treatment for VS. It can be administered in two settings: stereotactic radiosurgery (SRS),

which uses an irradiation of high dose in a single fraction, and fractionated radiotherapy or hypofractionated stereotactic radiotherapy (SRT). Despite a size of 2.5 being considered the limit for indications for radiotherapy, SRS is usually for cases of small to medium-sized VS, whereas SRT should be suggested in the case of larger VS. The aim of this treatment is to stop the tumor growth, with little damage to the nearby structures. Depending on the radiation dose, hearing function in these patients tends to decline over a long term. Nevertheless, in short-term postprocedural phase hearing performance often remains preserved [5–8]. Surgical approaches to VS can be divided into procedures with the attempt to preserve hearing, as with hearing-preserving surgery, and procedures with no hearing preservation attempt, like in translabyrinthine approach. The therapeutic choice is related to patient and tumor features, as well as surgeon preference [9–11]. The translabyrinthine approach reaches the internal auditory canal (IAC) through a presigmoid mastoidectomy and labyrinthectomy, with exposure of the sigmoid sinus and the retrosigmoid dura. This results in profound hearing loss. This approach is usually performed in patients with preoperative ipsilateral deafness or non-serviceable hearing [12]. HPS is represented by different surgical approaches, such as the "retrosigmoid" and the "middle fossa" [13–16]. It is usually performed in patients with small tumor size and functional preoperative hearing (assessed with tonal−vocal examination and auditory brainstem response ABR examination) [10]. Active treatment with HPS can offer a better chance for long-term hearing preservation than observation, and provides better rehabilitation chances with hearing aids or cochlear implants [17].

Hearing loss in VS can be classified as iatrogenic and tumorigenic. Iatrogenic hearing loss is correlated with the treatment of either surgery or radiotherapy. When a hearing preservation attempt fails, surgery provides an immediate solution for hearing treatment, while in radiotherapy, hearing loss takes longer before manifesting [18]. Tumorigenic hearing impairment is due to the natural history of the disease [19,20] and has multiple factors whose role remains to be defined. Among them, the following can be considered: growth patterns of tumors (no correlation was found by Gan et al., 2021 [11]), pathological alteration in the inner ear, aberrant inflammatory response, gene mutation, aberrant DNA methylation, and other [21–23] mechanisms that are still unknown.

Hearing loss is the most frequently reported symptom associated with VS. The typical form of hearing affliction caused by VS is a slowly progressive, high-frequency, unilateral, and asymmetric sensorineural hearing loss (SNHL) [24,25]. Less frequent are low-frequency losses or flat audiograms. Loss of speech discrimination is often associated with pure-tone hearing loss with worse hearing-in-noise performances. These features are typical in retrocochlear hearing loss. Indeed, VS causes multifactorial alterations within IAC, affecting CN function. In addition, the cochlea can be damaged during the progression of the disease [26]. Further evidence of this pathogenetic process be found in audiological tests such as ABR measurements [27].

When hearing preservation surgery is performed with successful outcome, the quality of the preserved hearing can be similar pre-treatment conditions, or it can deteriorate to various degrees. In the case of partial hearing loss, various types of hearing rehabilitation devices can be useful. Auditory rehabilitation should be considered in these patients, especially in those who have associated contralateral hearing impairment.

The available rehabilitation strategies for VS patients are conventional hearing aids, contralateral routing of signal (CROS) hearing aids, bilateral contralateral routing of signal (BiCROS) hearing aids [28], and bone conduction hearing aids (BAHA). Implantable electrical stimulation in complete deafness is provided through cochlear implants (CI) and auditory brainstem implants [29].

This paper is focused on the current knowledge about the use of hearing rehabilitation devices, such as CROS, BiCROS, and BAHA hearing aids, for both the condition of single-side deafness and hearing impairment in the VS-affected ear.

## 2. Materials and Methods

PubMed, Embase, and Scopus were systematically screened from January 2012 up to July 2022 using the following free term search: "hearing aid" OR "CROS" OR "BiCROS" OR "baha" AND "vestibular schwannoma". The literature search was independently performed by three authors (V.M.D.P.F., F.S., and D.B.). All the retrieved publications were evaluated to identify the most relevant ones. Duplications or aggregations of pre-existing data were excluded; only articles in English and Spanish were included. The reference lists of selected articles were also analyzed to identify additional studies.

## 3. Results and Discussion

After a review of the available literature, 283 articles were identified. Twenty-nine studies were considered for their relevance and matched to the topic of the present narrative review. The list of analyzed studies can be found in Tables 1 and 2.

**Table 1.** Review of the literature for conventional hearing aid rehabilitation in VS-related hearing loss.

| First Author | Year | Type | Title |
|---|---|---|---|
| Carlson ML [30] | 2012 | Case series | Cochlear implantation in patients with neurofibromatosis type 2: variables affecting auditory performance. |
| Drusin M [31] | 2020 | Observational | Trends in hearing rehabilitation use among vestibular schwannoma patients. |
| Fayad J [29] | 2010 | Review | Hearing preservation and rehabilitation in vestibular schwannoma surgery. |
| Jia H [32] | 2018 | Review | Neurofibromatosis type 2: hearing preservation and rehabilitation. |
| Johnson EW [33] | 1968 | Observational | Auditory findings in 200 cases of acoustic neuromas. |
| Macielak RJ [34] | 2021 | Observational | Hearing status and aural rehabilitative profile of 878 patients with sporadic vestibular schwannoma. |
| Meyer TA [35] | 2006 | Observational | Small acoustic neuromas: surgical outcomes versus observation or radiation. |
| Picou EM [36] | 2014 | Observational | Potential benefits and limitations of three types of directional processing in hearing aids. |
| Reffet K [37] | 2018 | Observational | Hearing aids in patients with vestibular schwannoma: Interest of the auditory brainstem responses. |
| Samii M [38] | 1997 | Observational | Management of 1000 vestibular schwannomas (acoustic neuromas): hearing function in 1000 tumor resections. |
| Snapp H [39] | 2012 | Review | Habilitation of auditory and vestibular dysfunction. |
| Totten DJ [40] | 2021 | Observational | Management of vestibular dysfunction and hearing loss in intralabirinthine schwannomas. |
| Woodson EA [41] | 2010 | Observational | Long-term hearing preservation after microsurgical excision of vestibular schwannoma. |

**Table 2.** Review of the literature for CROS and BAHA rehabilitation in VS-related hearing loss.

| First Author | Year | Type | Title |
|---|---|---|---|
| Andersen HT [42] | 2006 | Observational | Unilateral deafness after acoustic neuroma surgery: subjective hearing handicap and the effect of the bone-anchored hearing aid. |
| Bouček J [43] | 2017 | Observational | Baha implant as a hearing solution for single-sided deafness after retrosigmoid approach for the vestibular schwannoma: audiological results. |
| Clemis JD [44] | 1981 | Observational | The contralateral ear in acoustic tumors and hearing conservation. |
| Finbow J [45] | 2015 | Observational | A comparison between wireless CROS and bone-anchored hearing devices for single-sided deafness: a pilot study. |
| Hill SL [46] | 2006 | Observational | Assessment of patient satisfaction with various configurations of digital CROS and BiCROS hearing aids. |
| Lin LM [47] | 2006 | Observational | Amplification in the rehabilitation of unilateral deafness: speech in noise and directional hearing effects with bone-anchored hearing and contralateral routing of signal amplification. |
| Lotterman SH [48] | 1971 | Observational | Examination of the CROS type hearing aid. |
| Niparko JK [49] | 2003 | Observational | Comparison of the bone anchored hearing aid implantable hearing device with contralateral routing of offside signal amplification in the rehabilitation of unilateral deafness. |
| Ryu NG [50] | 2015 | Observational | Clinical effectiveness of wireless CROS (contralateral routing of offside signals) hearing aids. |
| Siau D [51] | 2015 | Observational | Bone-anchored hearing aids and unilateral sensorineural hearing loss: why do patients reject them? |
| Snapp HA [52] | 2017 | Observational | Effectiveness in rehabilitation of current wireless CROS technology in experienced bone-anchored implant users. |
| Snapp HA [53] | 2017 | Observational | Comparison of speech-in-noise and localization benefits in unilateral hearing loss subjects using contralateral routing of signal hearing aids or bone-anchored implants. |
| Snapp HA [54] | 2019 | Review | Nonsurgical management of single-sided deafness: contralateral routing of signal. |
| Vermiglio A [55] | 1998 | Observational | Development of a virtual test of sound localization: the source azimuth identification in noise test. |
| Wazen JJ [56] | 2003 | Observational | Transcranial contralateral cochlear stimulation in unilateral deafness. |

### 3.1. Assessment of Hearing Loss Candidacy to Hearing Rehabilitation with Hearing Aids

Pure tone audiometry and speech audiometry are the gold standard exams for the assessment of hearing function and the benefit of a rehabilitation with hearing aids. The

speech discrimination score (SDS), more than PTA, is a useful predictor of successful rehabilitation with hearing aids [35,41]. ABR can predict the auditory performance following auditory rehabilitation with HA in unilateral VS patients [37].

There is a shortage in studies regarding the degree of VS-related hearing loss that could benefit from hearing rehabilitation, and there are no specific indications for patients affected by VS regarding this topic. It has been described that patients using any type of hearing aid device most frequently have class C−D hearing (AAO-HNS: American Academy of Otolaryngology, Head and Neck Surgery). As expected, mild−moderate hearing loss with a good word-recognition score is considered the best condition for successful rehabilitation with hearing aids, which is a frequent condition in the HPS outcome [31]. Hearing aids are useful when word discrimination in the involved ear is at least 50%. If the level is lower, the amplified sound could appear to be distorted, resulting in poor hearing rehabilitation and failure of the auditory rehabilitation system [29]. Indeed, patients with a low speech discrimination suffer a deficit in hearing usefulness. In these cases, even if the pure-tone threshold is preserved, the possible overall result (in terms of speech discrimination) can be poor due to the worse ear interference on the better ear, resulting in a sort of masking effect [33]. On the other hand, good speech discrimination (70% or more speech discrimination, pure tone hearing within 30 dB) provides the patients useful hearing, or at least allows the patient to be rehabilitated with hearing aids [19,38]. Regarding patients with worse hearing conditions, a preference to avoid using hearing rehabilitation devices has been reported. For patients who did not find satisfactory rehabilitation with conventional hearing aids, alternative amplification methods could be suggested. These other options are represented by frequency-compression and frequency- transposition hearing aids. It is not easy to assess the possible benefit of these methods, considering the paucity of the available literature [39].

Few papers have investigated the rehabilitation of hearing loss in the less frequent intralabyrinthine schwannomas, describing poor results with different types of devices, such as conventional and CROS hearing aids [40]. Considering the features of hearing loss caused by a cochlear or retrocochlear disease as the VS of the VIII nerve, hearing rehabilitation has shown different results and clear indications are still lacking. A patient-centered approach, based on the specific audiologic, social, and psychological characteristics of the person, is currently the best strategy to assess the handicap of the patient and to drive the choice for the auditory rehabilitation.

### 3.2. Conventional Hearing Aids

The conventional hearing aid (air-conduction type) seems to be the most used type of device among VS hearing impaired patients [31]. This kind of rehabilitation is a possible choice for patients for preserving a residual hearing in the impaired ear. It could be suggested for those undergoing an observational follow-up, or those who received surgery with a hearing preservation approach. As previously stated, the results obtained with auditory rehabilitation in VS patients can be unsatisfactory, because of the mismatch between the tonal threshold and the speech intelligibility capacities typical of this disease. This gap is known to be relevant in retrocochlear hearing loss, such as VS hearing impairment. It is characterized by distortion, a condition that makes the amplification of sounds in these patients even more difficult [33]. Nowadays, hearing aids are rarely recommended for non-operated patients by specialists. As described in Reffet et al., 2018, the motivation can be found in specialists focusing on VS management and because of the limited understanding of auditory prognostic factors. Reffet used ABR, aiming to predict the results of auditory rehabilitation, as this evaluation can assess the integrity of the auditory pathway from CN to the brainstem. Moreover, Reffet, using the Glasgow Benefit Inventory (GBI) questionnaire, found an improvement in quality of life in hearing impaired VS patients rehabilitated with hearing aids. On the other hand, the researchers stated that more specific questionnaires should be administered [37].

A limited use of hearing aids among VS patients has been described in the literature. This finding is probably due to various factors. Among specialists, there is a general lack of knowledge regarding the available rehabilitative options. Many patients are not properly informed regarding these rehabilitation alternatives. Regular and repeated counseling is recommended to increase hearing rehabilitation after VS treatment [31].

Another factor influencing the use of hearing aids in VS patients is a limited perception of handicap. Most patients with unilateral hearing loss become used to their condition and prefer to avoid further rehabilitations, even in the case of a high degree of hearing loss. This can be explained by the fact that hearing aids, which are visible and clearly detectable on the patient, manifest the disability, whereas single-side hearing loss can be compensated by the contralateral ear, although at the price of a disabling of any progressive hearing fatigue. Other factors affecting the use of hearing aids may include the economic cost, the difficulty of use, and the initial limited benefit provided by the current hearing rehabilitation devices [34].

Hearing aids are considered a viable rehabilitation for those NF2 patients with serviceable hearing. These devices are mostly suggested for moderate−severe hearing loss as a temporary treatment. The type of hearing impairment of this illness, such as the retrocochlear damage, the decreased speech intelligibility, and the distortion, drive some patients to choose not to use aids [32]. Hearing aids are indeed considered inadequate for NF2 patients with deteriorated hearing [30].

### 3.3. Unilateral Deafness

Despite the presence of normal hearing function in one ear, verbal communication difficulties are quite common in unilaterally deaf patients. The unilateral deafness is characterized by two main kinds of impairment. First, the difficulty to detect sounds that come from the deaf side, due to the shadow effect of the head. As a result, the patient needs to turn the head to the sound source, to perceive sounds with the normal ear. Secondly, there is a loss of those binaural processes that usually improve the signal-to-noise ratio (SNR) and enable hearing localization [50,54]. Both consequences can lead to a reduced quality of life and a limited awareness of possible risks in daily life activities (driving a car, riding a bike, crossing the street in traffic, and others). On the other hand, CROS-Aid and BAHA are reported to be beneficial for patients with unilateral single-sided deafness. These types of devices collect the sound on the affected side and transfer the information to the contralateral ear to be processed [29].

### 3.4. Contralateral Routing of Signal (CROS)

The authors noticed a deficit in studies regarding the use of CROS hearing aids in patients affected by VS. Introduced for the first time in 1965 from Hartford and Barry, the CROS hearing aid is a traditional recommendation for unilateral deafness rehabilitation. It consists of two devices: a microphone placed in the deaf side that routes sound to a receiver positioned in the serviceable ear. This strategy mimics binaural sound processing and expands the area of hearing. In the literature, it has been reported that non-surgical cross-hearing devices were able to enhance the sound-to-noise ratio in noisy and reverberant spaces, especially when speech signal and background noise were spatially separated [50]. These types of hearing aids are not suitable in some cases: external ear canal or middle ear abnormalities or active pathologies (atresia or chronic suppurative otitis media). These conditions remain quite rare and CROS can be an option in most patients. The main reasons for poor compliance regarding these devices are aesthetic preferences, poor battery life, electromagnetic interferences (because of the radio frequency transmission), timing delays, and distortion [46]. Discomfort from the occlusion of the external auditory canal in the healthy side has also been reported. In some situations, these devices could also result in a degradation of speech intelligibility [48]. New models have wireless streaming with no more audible delays and virtually no interference [36]. Moreover, they have smaller sizes and a longer battery life. Environment recognition guarantees better noise reduction. New

CROS aids have been developed using an open-type hearing instrument in the normal ear, to reduce the sensation of occlusion of the normal ear. Nevertheless, many patients do not tolerate wearing a device in the normal-hearing ear [29,56].

Unfortunately, there is a deficit in the current literature regarding studies about the use of CROS hearing aids in VS patients. Consequently, the CROS hearing aid indication can be derived only from diseases with similar clinical features. The natural history of VS as well as surgery or radiotherapy can sometimes cause profound hearing loss. All of these conditions may lead to deafness, making these patients a possible target for this kind of rehabilitation [29]. Nevertheless, only future studies regarding CROS use in VS patients will be able to confirm this deduction.

BiCROS is like CROS, but it also rehabilitates contralateral non-VS hearing. A microphone on the deaf side provides an amplified sound to the better (but still impaired) hearing ear. It represents a useful device for SS-deaf people with a contralateral mild−moderate associated hearing loss [44].

### 3.5. Bone Anchored Hearing Aids (BAHA)

BAHA aids exploit transcranial bone conduction. The sound is routed to the better ear from a microphone, or a processor connected to an implant on the affected side, osseointegrated in the skull [45]. Hearing rehabilitation effectiveness with BAHA in VS patients is still controversial [43,51].

Few studies were conducted on BAHA hearing rehabilitation in patients with single-sided deafness after retrosigmoid removal of vestibular schwannoma. Boucek et al. studied the audiological results in 16 patients who accepted BAHA implantation. They found a significant improvement in sentence discrimination in the 6-week (64.0%) and 1-year (74.6%) interval at follow-up. This improvement was noticed in situations where sentences were coming from the side of the non-hearing ear, with −5 dB SNR contralateral noise [42].

Another study evaluated the subjective hearing handicap in patients with unilateral deafness after VS surgery and the effect of the BAHA on test band. Twenty-six patients attended the BAHA testing session, and they had a median improvement in the discrimination score of 15%. However, only half of them agreed to BAHA implantation after the test [49].

Many studies comparing CROS and BAHA technologies are available, but unfortunately none are specifically focused on VS patients. Some studies reported better speech identification with interfering noise with BAHA aids and a better acceptance from patients [47,53]. Other studies showed no differences in objective or subjective outcome with CROS technology compared with BAHA implants [45,52,55].

### 3.6. Assessment of Outcome in CROS and BAHA Rehabilitation of Unilateral Deafness

Severe to profound single-sided deafness reduces speech perception in noisy environments, especially when the sound arrives at the affected side and noise masks the better-hearing ear. Therefore, patients suitable for CROS and BAHA devices should be evaluated using speech-in-noise tests [47]. On the other hand, the differences are less appreciable in silent conditions. The benefit of CROS and BAHA should be assessed measuring SDS in the presence of competing noise. Testing should be performed with noise directed towards the better-hearing ear and speech stimuli towards the affected ear.

Questionnaires could also help evaluate the subjective benefit of using the devices. Lin et al. used the Abbreviated Profile of Hearing Aid Benefit (APHAB) to evaluate the effectiveness of hearing aids for speech understanding in everyday listening in patients using CROS [53].

As for the sound source localization assessment, the Source Azimuth Identification in Noise Test (SAINT) [47,53] is one of the most used instruments. This test measures the patient's ability to localize a sound source in the horizontal plane, in quiet and in noise, using auditory cues in a closed set of source locations. In the study by Lin et al., four different test stimuli were presented from the lateralized speaker locations on the array,

two stimuli were presented in a quiet background and two in a noisy background. Before localization testing, the detection threshold for each stimulation was determined in quiet and in the presence of background noise. Subjects were seated in front of the speaker array and responded after each stimulus, which was presented 10 dB above their respective threshold, by pointing to the perceived location of the sound source, without turning their head towards the stimulus. Responses were scored correct when the left versus right location of the sound source was identified [53]. Unfortunately, literature regarding the assessment of benefit in CROS or BiCROS devices is still lacking. Only future studies will be able to suggest if these patients may take advantage of these new technologies.

## 4. Conclusions

Vestibular-schwannoma-related hearing loss is a consequence of the natural history of the disease, or it can be the consequence of its therapy. Hearing devices are an available and useful tool for rehabilitating VS patients' hearing loss, both in untreated cases and in cases of patients who underwent surgery or radiotherapy. The impact on the entire hearing function is strictly influenced by the contralateral ear. Hearing preservation surgery and hearing rehabilitation with a cochlear implant for deafness has achieved a prominent role when planning the therapy, especially in small tumors, whereas rehabilitation with hearing-aid devices in VS patients is frequently neglected by clinicians. Nevertheless, it should be evaluated in relation to the degree of ipsilateral hearing loss and the degree of SDS in the affected side. Despite a lack of clear indications about the choice of the specific rehabilitation, the most frequent is amplification with conventional hearing aids, when serviceable hearing is still present. CROS and BAHA systems can be a treatment option in VS-related unilateral deafness. An accurate evaluation of outcome is necessary to detect the benefit of these devices. Cochlear implantation and auditory brainstem implantation are alternative surgical strategies that can be considered when CN has or has not been preserved after tumor removal, respectively. While specific studies on hearing rehabilitation have been conducted on cochlear implants after VS resection and on ABI rehabilitation, few studies have been performed to assess the audiological benefit with hearing aids (conventional, or CROS and BAHA). These strategies can improve the quality of life of these patients. Any choice should be evaluated in relation to the degree of hearing loss, the state of the contralateral hearing, the patient's personal willingness, and the feasibility of a good assessment of candidacy, as well as an evaluation of the outcome by specialized audiologists.

**Author Contributions:** Conceptualization, E.Z.; methodology, D.B. and F.S.; software, V.M.D.P.F.; validation, D.B. and F.S.; formal analysis, F.S. and V.M.D.P.F.; investigation, F.S. and V.M.D.P.F.; data curation, V.M.D.P.F.; writing—original draft preparation, F.S., G.D.L., M.C. and V.M.D.P.F.; writing—review and editing, D.B., C.d.F., F.S., E.Z., P.T. and V.M.D.P.F.; visualization, D.B. and V.M.D.P.F.; supervision, C.d.F., E.Z. and P.T.; project administration, D.B., E.Z. and P.T. All authors have read and agreed to the published version of the manuscript.

**Funding:** This research received no external funding.

**Conflicts of Interest:** The authors declare no conflict of interest.

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
