# Peer review of "Hearing Aid in Vestibular-Schwannoma-Related Hearing Loss: A Review"

_audiolres, doi:10.3390/audiolres13040054_

Round 1

Reviewer 1 Report

Many thanks for the opportunity of revising this interesting manuscript.

I only have some minor points:

- I couldn't find the Prisma diagram, with the related inclusion / exclusion criteria. I'd add the diagram among the figures / tables, and also the relative criteria in the text of the methods section, in oder to clear the readers this point.

- Table 1 line 192: I suppose it should be Table 2.

Author Response

Dear reviewer,

Thanks for your comments.

We made some changes in our work. We emphasized that our paper is a narrative review, therefore you will not find the PRISMA diagram.

We hope you'll find our reviewed work interesting.

Reviewer 2 Report

Authors intend this manuscript to be a Systematic Review of the hearing rehabilitation options for patients diagnose with VS leading to iatrogenic versus tumorigenic hearing loss. The authors, appropriately conclude that conventional hearing aids, CROS aids, BICROS aids, BAHA, cochlear implants, and ABI are potential options depending on the characteristics of the patient. However, many of these options are not offered to VS patients appropriately.

There are many details that are missing in this manuscript to label it as a systematic review.  Based on how it is written, it looks more like an overall review of the literature more than a systematic review approach. If this manuscript really intends to be a systematic review please address the following:

1.     Specify the inclusion and exclusion criteria

2.     Was an appraisal performed to those studies included in the review? If not, please explain why.

3.     Was an assessment for the risk of bias performed? If not, please explain why.

4.     Please add the PRISMA flowchart to the manuscript.

5.     Lines 36-37. Please rephrase. Translabyrinthine approach might be more common in certain institutions and not others. Please look more into this. Retrosigmoid approach has increased in popularity even if the patient does not qualify for HPS.  

6.     Line 228, change VZ for VS.

I agree with the fact that this topic is not well reviewed in the literature and it would be beneficial to have either a systematic review or a review of the literature on this topic, mainly discussing the use of conventional hearing aids, CROS aids, BICROS aids, and BAHA in VS patients.

Author Response

Dear reviewer,

Thank you for your interesting comments.

We made some changes in our paper. We emphasized that our work is a narrative review. Therefore, you will not find a PRISMA diagram.

We included in our review all the studies on hearing aids in VS available on main databases. The amount of these studies is particularly limited, so we thought that an overall review on the topic could be interesting.

We agree with you on translabyrinthine and retrosigmoid approach popularity. We rephrased our statement.

We hope you'll find our reviewed work interesting.

Reviewer 3 Report

This is a review paper on hearing aid uses in VS patients. It would add some values to the research area. However, it does not provide a comprehensive review on the papers in related studies. It is rather a quick summary of some published papers. In my mind, it is better to go a little bit deeper in each paper and discuss more about their methodologies and results. For review, please refer to a specific paper and summarize the most important findings from the studies. The readers will have an overall picture of what have been done and what are the major findings in this area.

"lacks ..." was used too many time in the review paper.

When citing papers, all reference numbers must be placed before a period mark.

It has too many grammar and word usage errors. To give a few:

Line 34: missing "such" in "such as"

Line 39: MRI=Magnetic resonance imaging

Line 41: change "conduct" to "treatment plan"

Line 43: "consists in" to "uses"

Line 45: “indicated” to "considered"

Line 48: "decline long-term" to "decline over a long term"

Line 80: "some authors" to " some studies"

Line 93: " address the current knowledge" to "present a review on the current knowledge"

Line 104: "authors" to "papers"?

Line 109: "may be" to "can be"

Line 109  "table" to "Tables"

Line 118: " grade" to "degree"?

Line 122-123: This sentence is not clear

Line 125 "appear to be distorted "

Line 133: " it is recommended not to use"

Line 135:  "represented by" inappropriate words

Line 149: "preserving " to " to preserve"

Line 159: "ABR is able to" to " ABR can be use to"

Line 163: not a good sentence

Line 184: "pick the hearing cues" to "attend to the auditory cues"

Line 187: "a bike" to "riding a bike" to be consistent.

Line 195: "from" to "by"

Line 199: " receive hearing" to "receiving sound"

Lines 232 to 234: The sentence is too long.

Line 258: “Al" to "al."

Line 274: this claim is too strong.

Author Response

Dear Reviewer,

Thank you for your comments. 

This topic is not widely debated in literature, and no review is currently available. Considering the paucity of evidence, our purpose is to make an overall narrative review of the available studies. We claim that further researches are needed to prove if what is currently known about similar hearing illnesses can be confirmed even in the case of VS.

We made some corrections in our paper. We also reviewed the English language of the document, so we hope you'll find it more readable. 

Round 2

Reviewer 2 Report

Dear authors,

Thank you for considering the my comments. There is definitely a lot of unknowns in this subject and more studies should be targeted to study hearing rehabilitation options and outcomes in this population. This is definitely a good way to start understanding the options and the few studies related to this topic. 

Reviewer 3 Report

All my comments have been addressed.